# User Reputation on E-Commerce: Blockchain-Based Approaches

**Maria José Angélico Gonçalves** [1,*] , **Rui Humberto Pereira** [2,*] and **Marta Alexandra Guerra Magalhães Coelho** [1]

1   CEOS.PP, ISCAP, Polytechnic of Porto, S.Mamede Infesta, 4465-004 Porto, Portugal
2   CEOS.PP, ISCAP, University of Maia, 4475-690 Maia, Portugal
*   Correspondence: mjose@iscap.ipp.pt (M.J.A.G.); rhp@iscap.ipp.pt (R.H.P.)

**Abstract:** User trust is a fundamental issue in e-commerce. To address this problem, recommendation systems have been widely used in different application domains including social media healthcare, e-commerce, and others. In this paper, we present a systematic review of the literature in the area of blockchain-based reputation models and we discuss the obtained results, answering the initial research questions. These findings lead us to conclude that the existing systems are based on a trusted third party (TTP) to collect and store reputation data, which does not provide transparency on users' reputation scores. In the recent literature, on the one hand, blockchain-based reputation systems have been highlighted as possible solutions to effectively provide the necessary transparency, as well as effective identity management. On the other hand, new challenges are posed in terms of user privacy and performance, due to the specific characteristics of the blockchain. According to the literature, two major approaches have been proposed based on public and permissioned blockchains. Each approach applies adjusted models for calculating reputation scores. Despite the undoubted advantages added by a blockchain, the problem is only partially solved since there is no effective way to prevent blockchain oracles from feeding the chain with false, unfair, or biased data. In our future work, we intend to explore the two approaches discussed in the literature in order to propose a new blockchain-based model for deriving user reputation scores.

**Keywords:** reputation system; user reputation; blockchain; blockchain oracles; e-commerce; security; fraud

## 1. Introduction

Nowadays, e-commerce competes in many economic sectors, side by side with traditional commerce. Despite other issues, such as the need to feel and touch a product and shipping costs, lack of trust is one of the greatest barriers to massive adoption of e-commerce. Never before has this aspect been as relevant as it is today due to the acceleration of the digital transition, as a consequence of the global pandemic, in which we have witnessed an increase in the number of e-commerce transactions over the internet [1].

In the context of a commercial transaction, trust and reputation are distinct but related concepts. "Trust is the extent to which one party is willing to depend on something or somebody in a given situation with a feeling of relative security, even though negative consequences are possible" and "reputation is what is generally said or believed about a person's or thing's character or standing" [2].

The lack of trust is particularly relevant in B2C and C2C online business models, in which there is no prior relationship (i.e., trust) between the participants, as generally exists in the e-commerce B2B model or in traditional commerce, where a buyer can feel and touch a product. Recommendations and reputation scores have been used to address this problem by helping the user, usually, a buyer, become aware of the risk during a transaction. In general, all online marketplaces (e.g., Amazon and eBay) provide recommendation and reputation data about the user or the product, typically in the form of a score, in order to provide trust to buyers. Consequently, the trust which supports a buyer's decision to buy a product/service is generally based on the reputation of the product or/and the seller.

In this paper, we conduct a systematic literature review with the aim of contributing to the field of blockchain-based reputation systems by providing an overview of the existing proposals for offering trust to e-commerce users. It should be noted that this work is directed at user reputation, because we consider product reputation to be a distinct problem that is independent of user reputation, despite complementing the trust given to a buyer.

Our work focuses on finding the existing blockchain-based models that define the reputation calculation process and data storage so that they can provide fairness and transparency to reputation scores. Thus, we formulate the following question: *"What blockchain-based reputation systems exist to determine user reputation in e-commerce?"*

In order to answer this question, as mentioned above, we conduct a systematic literature review based on three databases, following the PRISMA methodology [3], which we discuss in this paper, and we conclude that reputation systems have been effectively studied over a long period of time.

The proposed systems found in the literature are based on models that involve several data sources, such as personal data, social interactions, commercial transactions, and feedback given by other users. As this involves sensitive data, risk mitigation techniques are required, such as cryptography and protection of the users' private data. Moreover, the model on which the calculation of users' reputations is based should be immune to possible attacks and frauds that misrepresent the results of the calculations.

Despite the reliability of the results provided by the proposed models, to determine the reputation score of a user, in general, they are based on a trusted third party (TTP), which does not give process the necessary transparency. In the recent literature, the authors have proposed blockchain-based systems to overcome the lack of transparency of the reputation score given to a user or product by these TTP-based systems. The blockchain approach, since it is an approach based on a distributed ledger, grants the immutability of the data and transparency about the reputation scores. However, the blockchain is not a silver bullet solution, since it can introduce new types of attack, such as a 51% attack, as well as performance issues and user privacy.

Next, we present some very basic concepts that introduce a non-expert reader into the e-commerce field, providing support to fully understand the remaining sections. In Section 3, we discuss the methodology applied to our systematic review, as well as how it was applied in our work, enabling us to select the most relevant literature to be discussed in Section 4. We conclude the paper by presenting our conclusions and pointing out the future direction of our work.

## 2. Theoretical Background

In this section, we present the main concepts and strategies adopted in e-commerce reputation systems based on blockchain technology.

### 2.1. E-Commerce

Digital transformation along with the widespread adoption of the internet and mobile technologies has resulted in the creation of global markets where buyers and sellers transact for goods and services using different physical and virtual network architectures for offering, creating, and delivering value [4]. E-commerce refers to the sale and purchase of goods and services over the internet, with the transfer of money and data to complete transactions [5]. E-commerce platforms, in addition to selling products, facilitate the discovery of product information, which allows price comparison and decision-making about a purchase and the seller [6].

A company's business model represents its core logic and strategic choices for creating and capturing value within a value network [7]. The e-commerce model includes commercial transactions between buyers and sellers; the main flows of products, services, information, and money; and the main benefits to participants.

There are several e-commerce business models, and more are being created every day [5]; however, despite the abundance of models, it is possible to identify the main models that have been developed for e-commerce. According to Aithal [8], all the major e-commerce business models fall under three key categories: B2B, business-to-business; B2C, business-to-consumer; and C2C, consumer-to-consumer. In B2B, the businesses transact with other businesses. In B2C, business transactions are carried out with individual consumers and include purchasing of retail goods, travel, and other types of online services and content. C2C provides a way for consumers to sell to each other, with the help of an online platform. According to Laudon [5], the most discussed type of e-commerce is B2C commerce.

However, a lack of trust is particularly relevant in B2C and C2C online business models, where there is no prior relationship between the participants, as is usually the case in the B2B e-commerce model or in traditional commerce, where the buyer may know the participants (buyer and seller). Over the past few years, recommendation models and reputation rating rankings have been used to address this problem, helping the user, usually a buyer or seller, to minimize transaction risk. In general, all online marketplaces provide recommendation and reputation data, typically in the form of a score, about the user or product, in order to gain the trust of users.

### 2.2. The Role of Trust

In this paper, user reputation is the main focus, however, the concepts of trust and risk are important issues. Reputation and trust (or trustworthiness) are commonly confused [9] and used as synonyms, even though their meanings are distinctly different. Josang et al. [2] defined trust as "the extent to which one party is willing to depend on something or somebody in a given situation with a feeling of relative security, even though negative consequences are possible". Risk is often taken in the hope of some gain or benefit. Therefore, risk can be viewed as a situation where the outcome of a transaction is important to a party, yet the probability of failure is not zero [2]. By integrating the two definitions one can conclude the following: The amount of risk a party may be willing to tolerate is directly proportional to the amount of trust it has in the other party.

### 2.3. Reputation Systems and Reputation Models

The main purpose of user reputation systems is to establish trust between unknown parties. Based on a reputation model, a reputation system enables the collection, aggregation, and distribution of data about an entity that can, in turn, be used to identify and predict the future actions of that entity. Using this data, e-commerce users can decide whom they will trust and to what degree. Reputation systems increase or decrease user ratings according to the information collected about the user. They can, therefore, give a positive score, leaving the user with a better ranking, or they can give a negative reputation to punish dishonest behavior. As a result, many online marketplace platforms have developed user reputation management systems that allow trading parties to submit a rating of the counterparty performed in a specific transaction, which is made available to all site visitors. A positive rating of a trading partner is likely to increase trust in the counterparty's performance.

According to Hoffman et al. [10], reputation models are composed of three fundamental dimensions: (1) formulation, (2) calculation, and (3) dissemination. In the formulation dimension, the mathematical basis and input types that feed the model are derived. In the calculation dimension, the algorithm of calculation produces a reputation score from the input data. The latter regards the mechanism that allows system participants to obtain the calculated reputation score.

Regarding the formulation dimension, the authors propose the following types: manual feedback, direct and indirect observations, and inferred data. Reputation systems quantitatively construct sellers' reputations by collecting feedback from buyers with whom the sellers have ever interacted, where feedback is usually presented by a rating that reflects the sellers' performances. The automatic sources are obtained automatically either

via direct or indirect observations. In direct observations, automatic sources of information result from data directly observed by an identity, such as the success or failure of an interaction, or the direct observations of cheating. In the case of peer-to-peer (P2P) networks, the measurement of resource utilization is done by neighbors. Information that is obtained second hand or is inferred from first-hand information is classified as indirect. Liu et al. [11] proposed the (3R) model that incorporated observations, based on a buyer's repurchase/product return behavior information, into the calculation dimension in order to mitigate the negative impact of biased ratings.

Hendrikx et al. [12] proposed two families of reputation systems, explicit and implicit reputation systems. According to the authors, on the one hand, explicit systems have models that follow the same aforementioned principles and dimensions, in which a formulation and calculation have been explicitly defined. On the other hand, the implicit systems are not implemented in network services, but conceptually there is a model of reputation. As examples, the authors give social networks, such as Facebook or LinkedIn, and the Google search engine. In the first case, trust/reputation is inferred through relationships, i.e., friends of friends, while in the search engines, reputation is determined by the number of links that point to the page, and where the links originate. These are examples of automatic observations and inferred data that can enrich the model formulation in the explicit reputation models.

In the next section, we discuss the formulation and calculation dimensions of the reputation models that are subject to fraud based on several types of attacks and combinations of them.

### 2.4. Common Vulnerabilities of Reputation Systems for E-Commerce

The rapid growth of e-commerce platforms and their extensive use and dependence on their reputation systems have led to various types of malicious behaviors and threats.

Dellarocas [13] identified two problems related to a major approach for deriving users' reputations based on feedback: (1) unfair ratings by buyers and (2) discriminatory seller behavior. In the first case, two types of fraud can occur: Ballot-stuffing fraud, an attack where members positively rate themselves on fake (unfair) transactions in order to inflate their reputation and bad-mouthing fraud as a result of an attack where members misclassify others to deflate their reputation. These attacks are orchestrated in a collusion of a seller, or a group, and buyers. In discriminatory behavior, the seller strategically provides good service to a group of users and bad service to others, in order to gain benefits from that asymmetry of product/service quality. According to Panagopoulos [14], the most common threats against reputation systems include ballot-stuffing, bad-mouthing, and traitor attacks. The traitor attack is a type of attack where members exploit their reputation by tricking others until their reputation dissolves.

The author also mentions that the reputation systems use a much larger population sample than in the past and this fact may cause greater measurement bias from users who do not leave feedback on their transactions.

Following a suggestion by Brian Zill, Douceur [15] proposed the term *Sybil* attack in the context of P2P networks. In this type of attack to reputation systems, an entity forges multiple identities in the system, using it in collusion as a means to increase its influence. Whitewashing is another vulnerability in the identity management scope. In this type of behavior, an e-commerce user with a bad reputation can easily create a new identity and continue his activity without any consequence of his past transactions.

The architecture of the reputation system has also been subject to criticism due to vulnerabilities when centralized or distributed. According to Zulfiqar et al. [16], central authorities can potentially filter, tamper, add, or reject product reviews based on their preference. Schaub et al. [17] pointed out the same issues since, potentially, a centralized system can be abused by the central authority. Dhakal and Cui [18] presented the same arguments, stating that the current centralized systems are silos and non-transparent in the review process. In addition to the lack of transparency, these isolated centralized systems

do not benefit from the reputation data of each other. Regarding the attacks based on the limitations of identity management, such as whitewashing and Sybil, Zeynalvand et al. [19] stated that it was hard to derive a robust model if users did not share information. However, in the case of decentralized systems, nodes could potentially manipulate the data users shared in the network, even if it was encrypted or signed. We consider these issues to be a major problem, regarding the lack of transparency of reputation data, as well as a major difficulty to derive a robust reputation model. These problems are our focus in the present paper and are discussed in the following sections.

In the literature, one can find several proposals for classifications and taxonomies of attacks on reputation systems [10,20–22]. In these previous works, the known types of attacks and limitations are discussed. However, we should notice that we clearly identify and distinguish two types of attack families. One family of attacks is focused on the vulnerabilities of the reputation model, briefly discussed in the present section, which regards the methodologic approach of calculating the reputation users' scores. The second type of attack family regards the vulnerabilities in the technological infrastructure, such as, at the network level (e.g., the use of obsoletes protocols, such as SSLv3 and man-in-the-middle attack) or due to outdated software, which contain security flaws that can be exploited by attackers. Another observation is that, in this second family, in general, the attacks come from outsiders while the attacks on reputation models, in the first family, come from users/nodes of the community. We claim that this latter family regards a distinct field of research, thus, it is out of the scope of the present work.

The above facts create uncertainty about the level of security and trust that traditional reputation systems based on feedback or other observations taken from the environment can provide to e-commerce platforms. Hence, there is a need to create improved reputation systems that can operate effectively on these platforms and are resilient against malicious attacks.

As previously mentioned, the present work is centered on the vulnerabilities of the reputation models, with a particular focus on architectural issues. We believe that the transparency of reputation data is a major means to achieve robust reputation models.

*2.5. Blockchain Technology*

Over time, reputation systems have been widely implemented in e-commerce applications [23]. As already mentioned, reputation information can be stored in a decentralized or centralized manner. Decentralized storage of information, that is, shared from one node to the others in a distributed system, has advantages; however, it also has several challenges, such as those discussed in the previous section, which allow for ballot-stuffing and bad-mouthing frauds. These drawbacks render reputation systems useless since they cannot ensure the integrity of trust ratings, prevent data manipulation, and provide reliable mechanisms to support effective user identity management.

According to the recent literature, blockchain-based approaches to reputation systems may be capable of addressing these problems. Blockchain is the primary technology for Bitcoin and other digital currencies [24]. Stakeholders such as developers, entrepreneurs, and technologists claim blockchain technology has the potential to reconfigure the contemporary economic, legal, political, and cultural landscape. A smart contract removes the need to build trust between individuals and organizations through intermediaries such as lawyers, and social activities such as meetings, where actors get to know one another. Smart contracts build the transactional relationship of a contract into a technical code that is executed automatically.

As noted by Sherman [25], blockchains are a way of changing the way online reputation systems are managed. By integrating a proof-of-individuality framework into the verification system, a blockchain model protects against attacks and prevents scenarios such as spoofing, creating multiple identities, and manipulating scores. This technology has driven researchers to make new advances in continuing to perform processing efficiently. However, we should note that some issues cannot be directly addressed by the blockchain

paradigm, such as false and unfair ratings, discriminatory behaviors, and the "bias towards positive ratings" [2] explained as positive ratings as an exchange of courtesies or given in the hope of getting a positive rating in return, and negative ratings that are avoided because of fear of retaliation from another party. These problems are already known in the literature as the blockchain oracle problem [26–29].

As more and more people rely on online services and communities, user reputation systems will continue to play an increasingly important role in facilitating their interactions. It is already clear that online services can play a profound role in business. Therefore, implementing robust user reputation systems is crucial.

## 3. Materials and Methods

We systematically reviewed the scientific literature on blockchain technology in user reputation systems. On the one hand, we sought to identify the main common vulnerabilities of reputation systems for e-commerce and, on the other hand, to know the architectures of blockchain-based reputation systems that mitigate these vulnerabilities. Our search timeline included the years 2010–2021.

Systematic reviews are a form of meta-analysis designed to collect, investigate, and summarize what is known and what is not known about a "specific practice-related question" [30]. Systematic reviews are used across a broad range of disciplines, and qualitative studies have established a place for themselves within the methodology, as evidenced by initiatives such as the Cochrane Qualitative Methods Group [31], and textbooks such as *Systematic Reviews in the Social Sciences* [32] and *An Introduction to Systematic Reviews* [33].

In this study, in addition to following the primary objectives of the systematic review as defined by J. Frizzo-Barker et al. outside PRISMA [3], we also substantiate the results obtained, as outlined in Section 4, with a literature review, presenting theoretical perspectives and innovations from leading authors in the field. Systematic reviews have several positive features for social sciences [32,33].

There are some steps to conducting a systematic review. In the first step of the methodology, several research questions are defined to be answered based on the literature review results. In the second step, a protocol is defined to support the evaluation of the scientific studies that are relevant to the study. The last step involves the process of answering the research questions initially raised (in the first step), based on the scientific papers identified as relevant (in the second step). Figure 1 presents the steps followed by the adopted methodology.

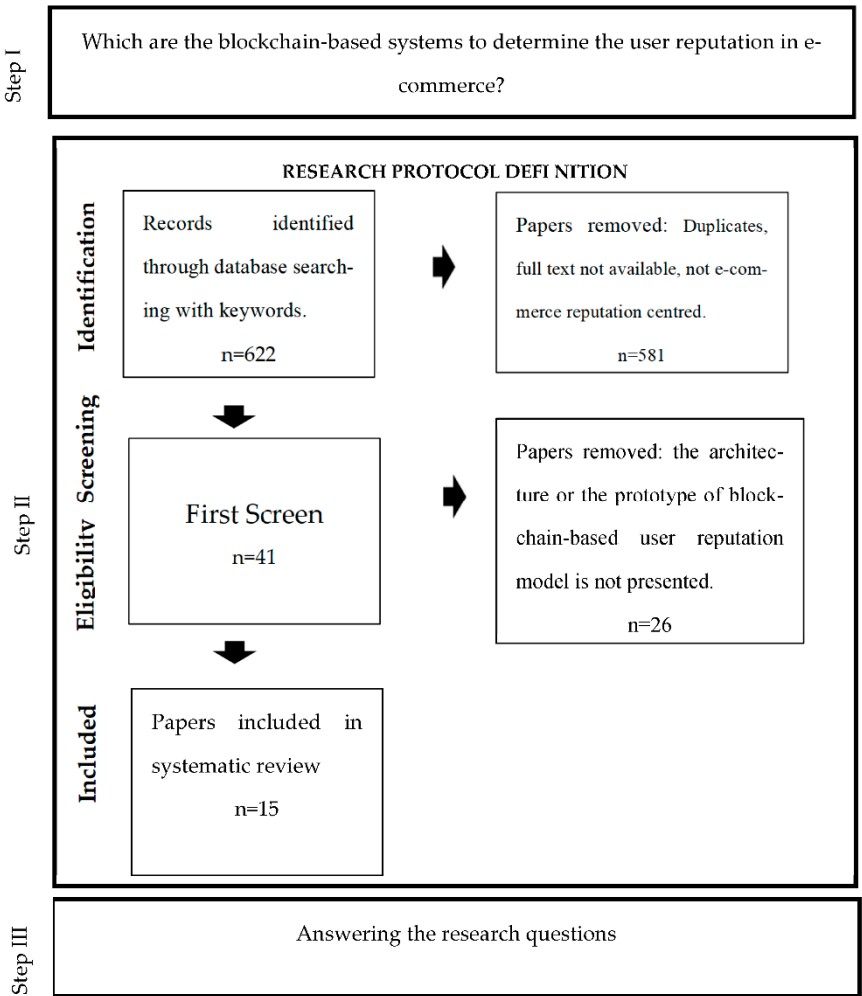

**Figure 1.** Steps followed in the systematic review.

### 3.1. Definition of Research Questions

The first step of the adopted methodology is related to the definition of the research questions of this study. The main research question intends to raise the state of the art concerning our study characteristics (traced in the Introduction): *"What blockchain-based reputation systems exist to determine user reputation in e-commerce?"* This question was subdivided into the following questions:

*Do current architectures provide guarantees of transparency and immunity to attacks and fraud?*
*How are these limitations being addressed in blockchain-based architectures?*
*What are the proposed blockchain-based reputation systems architectures?*

After the definition of the research questions, the second step was related to the selection of the empirical data to be analyzed.

### 3.2. Conducting the Search

In step two, i.e., data collection, we developed our search protocol, which outlined the methods used to carry out the systematic review. This process was designed to reduce researcher bias since a systematic review is often a collaborative effort [34].

This step was decomposed into four phases (following the PRISMA statement approach [3], see Figure 1, research protocol definition).

### 3.2.1. Identification

We started our research work by searching the scientific literature on blockchain-based reputation systems. The search was conducted in the two main databases where articles with the highest impact factor in the scientific area in question are located, Web of Science (WoS) and SCOPUS. The use of these databases makes the article more robust, as it covers more articles of greater academic importance. Additionally, we conducted grey literature searches (Google Scholar and Scholars' web pages) to complement and update the results. The search was conducted during the month of May 2021.

Using the WoS and SCOPUS databases, we searched for papers that included TOPIC: (trust AND electronic commerce AND blockchain) OR TOPIC: (online identity AND commerce AND blockchain) OR TOPIC: (online trust AND blockchain) in their title, abstract, or keywords. This search resulted in 101 articles selected from WoS and 501 articles selected from Scopus. The lists were exported to excel for further analysis and the following fields were chosen: authors, title, year, link, abstract, and keywords. Additionally, in Google Scholar, we performed a manual search with the same terms. Then, we selected the articles from the first 3 pages of the results (20 papers), using the same method; we extracted the fields referred to above and filled in an excel sheet.

### 3.2.2. Screening and Selection of Relevant Articles

Next, we evaluated the articles based on the inclusion criteria to determine their relevance to our study. An article had to include the search terms as the core technology under analysis. This was typically evidenced by its emphasis on the title, abstract, and keywords. We selected only academic peer-reviewed journal papers and conference proceedings and excluded others, namely: (a) papers without full availability, (b) papers not available in English, (c) duplicate articles, and (d) papers that did not discuss the reputation system, or its models, from technical, engineering, or computing science perspectives. The identification and inclusion process of our systematic review is presented in Figure 1. Our initial search was carried out in May 2021 and yielded 622 articles. Once we eliminated duplicates, entries without full-text availability in English, and papers that were not e-commerce-reputation centered, we were left with a population of 581 papers. Next, our research team, including a professor, two doctoral students, and a master's student, who reviewed this collection of articles for relevancy. In the first round of our inclusion process, we assessed the articles for their relevance based on title, abstract, and keywords. This process led to the selection of 41 articles. Any articles we did not agree upon, were also excluded. In the next round of revisions, we assessed the articles based on the full paper. We eliminated 26 papers that only concerned the architecture or where the prototype of the blockchain-based user reputation model was not presented. Thus, we identified 15 relevant articles for analysis, which are listed in Table 1.

**Table 1.** Papers included in the systematic review.

| Reference | Title | Contribution |
|:---:|:---:|:---:|
| [19] | A Blockchain-Enabled Quantitative Approach to Trust and Reputation Management with Sparse Evidence | Proposal and evaluation of a trust and reputation management (TRM) framework based on a mathematical model. |
| [18] | DTrust: A Decentralized Reputation System for E-commerce Marketplaces | Design, implementation, and evaluation of a decentralized and sharable reputation system for e-commerce, based on the Ethereum blockchain and IPFS. |
| [16] | EthReview: An Ethereum-based Product Review System for Mitigating Rating Frauds | Ethereum blockchain-based P2P product review system for e-commerce. |

**Table 1.** *Cont.*

| Reference | Title | Contribution |
|---|---|---|
| [35] | Rep on the Block: A Next Generation Reputation System Based on a Blockchain | Proposal for a blockchain-based generalized reputation system that can be applied to multiple networks, such as e-commerce and P2P. |
| [36] | Rep on the Roll: A Peer-to-Peer Reputation System Based on a Rolling Blockchain | In addition to [35], the authors provide a possible solution to a fundamental issue, the scalability, in blockchain-based networks. |
| [37] | Anonymous and Verifiable Reputation System forE-commerce Platforms based on Blockchain | Proposal and evaluation of a prototype, named RepChain, based on the Ethereum test network for e-commerce. |
| [38] | Reptor: A Model for Deriving Trust and Reputation on a Blockchain-Based Electronic Payment System | Proposal and evaluation of a model, named Reptor, based on the Ethereum test network for e-commerce. |
| [39] | A Model for Deriving Trust and Reputation on a Blockchain-based e-Payment System | In addition to [38], the authors implement their model into the blockchain-based platform (http://nodehome.io, accessed on 15 May 2021). |
| [40] | Blockchain-Based Global Travel Review Framework | Set of guidelines for the development of a platform based on Ethereum for the tourism sector. |
| [17] | A Trustless Privacy-Preserving Reputation System | Proposal of a protocol and its evaluation in terms of security, malicious behavior, and robustness against generic attacks. |
| [41] | A Secure Personal-Data Trading System Based on Blockchain, Trust, and Reputation | Proposal of a model and its implementation as a prototype of a permissioned blockchain using the platform Hyperledger Fabric v2.0. |
| [42] | Decentralized Reputation System on a Permissioned Blockchain for E-Commerce Reviews | Proposal of a decentralized reputation system on a permissioned blockchain, token generation method, and prototype using Hyperledger Fabric, which allows retailers to establish reputations of products and, by extension, vendors, or manufacturers. |
| [43] | Anonymous Reputation System for IIoT-Enabled Retail Marketing Atop PoS Blockchain | Proposal of a reputation system for consumer–retailer channels, in the context of the Industrial Internet-of-Things (IIoT) ecosystems, and the implementation of a prototype using the Parity Ethereum. |
| [44] | Blockchain-Based Decentralized Reputation System in an E-commerce Environment | Proposal of a decentralized reputation system using the blockchain, IPFS, and smart contract technologies for e-commerce. The evaluation of the proposed system was based on a simulation using a testing framework for Ethereum. |

**Table 1.** *Cont.*

| Reference | Title | Contribution |
|:---:|:---:|:---:|
| [45] | Using Blockchain Technology To Improve Trust in E-Commerce Reviews | Proposal of the use of blockchain in order to place trust in the technology rather than the benevolence of a party. The author discusses his proposal in two areas: (1) Generation of the review blockchain and (2) access to the review blockchain. |

The third and last step of the methodology, answers the research questions raised initially which discussed in Section 4.

Supported by the papers selected, we present our discussion.

## 4. Presentation and Discussion of the Results

After analyzing the articles listed in Table 1, we found that they propose to solve the problem of user reputation management by applying a blockchain-based approach. Each article tends to embrace particular aspects of the problem, as well as focuses on different contexts: P2P networks and e-commerce. In order to summarize and support our discussion below, Table 2 presents a systematic summary the most relevant findings in the studied works. Regarding the structure of this table, the name given by the authors to the system/model is presented in the first column (if it is available), followed by the bibliographic references in the second column. In the third column, the target of the reputation system, products, and/or users (buyer and/or seller) are presented in the scope of e-commerce. In the fourth column, we present the major problems addressed by blockchain-based architecture. In the fifth column, the state of the proposal (reputation model, prototype, or production system), as well as the addressed challenges of the blockchain-based architecture are presented. In the last column, we present a list of additional observations that may enrich the model in terms of accuracy and immunity to the aforementioned attack types.

**Table 2.** Comparative analysis.

| Project | Reference | Target | Problems to Be Addressed by the BC | State/ Addressed Challenges | Additional Observations |
|:---:|:---:|:---:|:---:|:---:|:---:|
| EthReview * | [16] | Products | Rating frauds: bad-mouthing and ballot-stuffing + Lack of transparency due to a central authority | Prototype/ Ethereum | Endorsers + economic viability when rating |
|  | [40] | Products (Tourism) | Lack of transparency and incoherence in review scores, as a result of isolated databases and different practices | Guidelines/ tested on the Ethereum Ropsten test network/Reduce the amount of data in the blockchain, operations, and costs using IPFS | Community-driven model (voting and rewarding) |
| BEQA * | [19] | Users | Sybil and whitewashing attacks | Mathematical model | Economic viability when rating |
| DTrust * | [18] | Users and Products | Rating frauds: bad-mouthing and ballot-stuffing + Lack of transparency due to a central authority: data manipulation + the data is not shared | Prototype/ Ethereum/Reduce the costs using IPFS | Community-driven model (voting and rewarding) Financial incentive for the reviewers |

**Table 2.** *Cont.*

| Project | Reference | Target | Problems to Be Addressed by the BC | State/ Addressed Challenges | Additional Observations |
|---------|-----------|--------|-----------------------------------|----------------------------|------------------------|
| **Rep On block *** | [35] | | | | |
| **Rep on the Roll *** | [36] | P2P nodes and generalized for e-commerce users | Lack of transparency due to a central authority + Sybil attacks | Proposal/ Support high rate of transactions and the blockchain size | P2P—Remove human option from transaction + Financial penalties for dishonest users + Cost in entrance to the network |
| **RepChain *** | [37] | Users | Centralization: Lack of transparency due to a central authority and single point of failure + Isolation: Reputation data is not shared | Prototype/ Ethereum test network/ Rating privacy, identity privacy, and unlikability | None |
| **Reptor *** | [38] | | | | |
| | [39] | Users | Transactions and reputation data is prone to manipulation by malicious entities + Different evaluation criteria | Model and Prototype/Homenode/By means of a cache, reduce the latency of the queries to the blockchain | Human behavior, psychological factors, Time and difference weigh |
| | [17] | Users (sellers) | Possible abuses by the central authority, which makes the centralized systems trustless | Model/Reduction of the time to compute the reputation of a seller. Linking the transactions of a service provider | Economic model based on incentives and costs when interacting with the blockchain |
| | [41] | Data Sellers | In centralized systems, users lose control of personal data, payment of high fees, and signing of terms that often compromise privacy, and still be subject to data leaks + Attacks from malicious users | Prototype/Hyperledger/With major performance concerns | Adaptive ageing function. |
| | [42] | Products | Customer reviews and ratings are locked to the retailer's platform + Unclear metrics + Reputation data is not shared | Prototype/Hyperledger/ Without major performance concerns | Limits to the number of reviews given by a customer to a product and control of multiple reviews in save order |
| | [43] | Sellers (retailers) | Insufficient system transparency | Prototype/Ethereum Parity/off-chain rating token generation phase in order to reduce the on-chain storage and computation overhead | None |
| **BC-DR *** | [44] | Sellers, buyers and comments for the products | Centralized reputation systems might make errors or even engage in fraud and forgery: data modification and fake comments and ratings | Model evaluated on a testing framework for Ethereum/ The IPFS is used for storing data | Three weighting factors: (1) the transaction time, (2) transaction amount, and (3) the previous reputation scores of users. + Monetary incentive mechanism |
| | [45] | Products | Business could provide incentives to the rater or reviewer to provide fraudulent or biased reviews + The data and mechanisms are locked to the marketplace's own platform | Proposal | External services, such as IBM Watson, to grade the quality of the reviews + Some reviews expire + Minimal number of reviews to affect the trust |

* When the articles referred to the name of the project, the analysis was carried out by project. We found that there were several publications referring to one project at different stages.

Table 2 presents, very briefly, the main characteristics of each project.

### 4.1. Management of Reputation

Although some of the discussed proposals are not directly focused on user reputation, but on product reviews [16,40,42], we found very interesting ideas here, which may mitigate the aforementioned problems in the context of user reputation. Based on the same principle,

we also include, in this literature review, works in the field of user reputation in P2P networks. In these cases, according to the authors, their proposals [35,36] are generalizable to both scenarios: e-commerce and P2P networks.

### 4.2. Transparency

Within the selected literature, we find a common motivation in all works, i.e., the lack of transparency resulting from the centralization of data on a single platform, managed by an authority. According to the authors, the current reputation systems based on a central authority are prone to data manipulation and errors. As a solution to this problem, the authors propose a blockchain approach due to its immutability and tamper-proof characteristics.

Indeed, the blockchain paradigm ensures that all data stored in the chain is immutable. Therefore, the reputation data (calculated scores and all records of manual feedback, direct and indirect observations, and inferred data [10]) are preserved in the chain ensuring the desired transparency, and avoiding any type of data manipulation. However, from a blockchain point of view, the e-commerce platforms are oracles that feed them, which can potentially introduce false and/or unfair data [26–29], as well as biased reputation data [11,45]. This problem, also presented as a limitation of the smart contracts in [28], potentially compromises this desired transparency. As such, in terms of transparency, the blockchain-based approaches partially solve the problem, so additional means are needed in order to avoid low quality data from entering the chain.

Based on the literature, we state that this transparency in reputation systems mainly concerns data about the users' reputations. The lack of transparency enables malicious users to perpetrate ballot-stuffing and bad-mouthing frauds. Moreover, this problem is extended to the data about the user identities, because a noneffective identity management enables means to Sybil and whitewashing attacks. In the following two sections, we discuss these types of problems.

### 4.3. Effectiveness of the Reputation Scores Calculation—False, Unfair, and Biased Feedback

Regarding user reputation data, as discussed in Section 2.4, its main concerns are about the quality and honesty of the feedback in the sequence of a commercial transaction. In order to address this problem, there are several approaches proposed in the literature that take advantage of blockchain-based architecture.

The first one is the sharing of data reputation across the chain, thereby, solving the isolation problem [37] that makes platforms as data silos [18]. By breaking this isolation, platforms can share their data among themselves, increasing the total volume of available data and enabling their algorithms to become more efficient while deriving reputation scores, as well as more resistant to ballot-stuffing and bad-mouthing frauds [19,37]. Other approaches based on machine learning [45] also benefit from this volume of data. Despite the efficiency of the algorithms, in such a decentralized environment the same data are provided to users avoiding incoherent information about feedback when visiting distinct platforms. Additionally, the calculation of the reputation scores could be more transparent, even if each platform applies different methods and algorithms.

In [16–19,35,44], a financial model is proposed as a strategy to reduce fraud. The authors stated that one effective way to reduce ballot-stuffing and bad-mouthing frauds, is by making them economically unviable due to required costs. In their proposals, when submitting feedback in the sequence of a commercial transaction, the seller supports a cost, in general, the cost of the transaction in the public blockchain (e.g., Ethereum Gas). Additionally, honest users are rewarded. This approach is not exclusive to blockchain-based reputation systems, because it could also be applied in other distributed environments. However, a public blockchain provides an environment that enables a seamless integration of the reputation system and platforms, as well as in terms of identity management, as we discuss below.

In order to mitigate the fraud and bias in feedback, in [16,18,40] a community-driven approach based on endorsers and voting mechanisms, complemented with rewards, is

proposed. According to the authors, if users become motivated to classify and validate feedback from other users, the quality of the reputation data is significantly improved. Again, this approach is also not exclusive to blockchain-based reputation systems, because it could also be applied in other distributed environments in which e-commerce platforms share their data.

### 4.4. Effective Identity Management

As discussed in Section 2.4, Sybil and whitewashing are two problems that result from a lack of effective identity management. In [16,18,19,40,44], it is proposed to map the user on the e-commerce platform and its identifier on the blockchain. This way, the user is always the same in each platform, enabling the cross-analysis of its activity in all platforms, as well as its reputation score is coherent from one platform to another. Additionally, the aforementioned problems, Sybil and whitewashing, are also addressed because it is not easy to create new and false identities.

In the case of permissioned blockchains, their management is ensured by a consortium of platforms [41,42]. In order to be accepted in the consortium, a platform must follow the imposed rules. In terms of user management, the authors proposed a central authority for managing users, validating their identities, as well as certificate issuing.

Due to its properties, blockchain-based reputation systems enable new models based on economic viability and community-driven collaboration, as well as a means for better identity management, which reinforces the effectiveness of existing algorithms for deriving reputation scores. These new models do not solve the problem of data quality and lack of confidence in the blockchain oracles, but they are empowered by the immutability and tamper-proof properties provided by the blockchain.

### 4.5. Performance, Costs, and Security

Despite the advantages of the blockchain-based approaches in reputation systems, some challenges are posed in terms of performance, operation costs, and security.

In a blockchain network, all history is preserved from the "genesis block" until the last transaction in the last block. Despite the transparency enabled by this approach, it is very challenging in terms of performance, because in real world e-commerce scenarios, such as Amazon and eBay, there is a huge number of daily transactions, each one requiring stored data and operation costs in the blockchain (the public ones). Note that feedback from a product sale may imply more than one transaction in the blockchain and these costs are variable due to the fluctuation of the cryptocurrency price in the market.

In the studied literature, we found four approaches for dealing with these two issues: (1) use of external storage systems [18,40,44], such as IPFS, in order to reduce the amount of data in the blockchain, as well as to reduce the costs of operation in the public blockchain; (2) cache mechanisms [38,39]; (3) modifications in the blockchain in order to reduce its size, and generation of a daily genesis block [36]; and (4) link the blocks of the same seller [17]. According to the authors, these approaches mitigate the performance issues and costs, but the impact on a real-world global blockchain network that interconnects platforms such as the ones referred to above is unclear.

A blockchain network is a public ledger to which all participants have read permissions over the data in the blocks. This requires additional mechanisms of cryptography, typically based on tokens, bling signatures, and asymmetric encryption, in order to protect users' privacy, for instance, avoiding retaliation to the user rating a product or seller, making users and their feedback unlikable. In the literature, there is much attention to this issue [43].

The blockchain was initially proposed using a POW consensus algorithm, which is a secure algorithm, despite its financial costs in the miner nodes [24]. In order to solve this problem, other consensus algorithms have been proposed and adopted, such as the PoS in the public Ethereum blockchain. Independent of the consensus algorithm, the blockchain is compromised only when the attacker controls 51% of the network.

*4.6. Research Outcomes*

In Section 3.1, we formulated our research questions. After presenting and discussing the selected works, we will now try to answer those questions.

*(i)    Do current architectures provide guarantees of transparency and immunity to attacks and fraud?*

According to the literature, the current reputation systems, centralized or decentralised, suffer from several limitations: (1) They are managed by one entity, which does not give all guaranties of transparency. (2) They are isolated systems that do not share the reputation data, therefore, a user or product may have distinct reputation scores in each platform. (3) The identity management is not effective, thus, enabling Sybil and whitewashing attacks. (4) Since the reputation data are not shared by the e-commerce platforms, it is harder to combat common fraud types, i.e., ballot-stuffing and bad-mouthing.

Based on these findings, our answer is no, the current architectures do not provide guarantees of transparency and immunity to attacks and fraud.

*(ii)    How are these limitations being addressed in blockchain-based architectures?*

As proposed in the literature, two major approaches exist based on: public blockchain networks [16,18,19,40,44] and permissioned networks in consortium [41,42]. The first approach is supported on economic viability, in which it is not economically viable to be a dishonest user, and honest users are rewarded. In the latter approach, the e-commerce platform is accepted in a consortium by agreeing with a set of rules. Typically, there is a central agency that manages the consortium, as well as a central management of user identities and digital certificates.

Regarding the vulnerabilities in identity management, the literature proposed two approaches: (1) costs in creating new identities and (2) a central authority. The approaches both intend to ensure that an entity has only one user on all platforms, thus, mitigating the Sybil and whitewashing attacks, in general, by linking the user in the e-commerce platform to his address in the blockchain network.

In short, in blockchain-based systems, the reputation model formulation could be more effective when deriving the reputation scores, mitigating the common types of fraud, i.e., ballot-stuffing and bad-mouthing frauds. By means of economic models that apply costs to operations and financial incentives such as rewards, or financial penalties to dishonest users, one could improve the model's performance. Community-driven approaches are also proposed, in which users validate the feedback from other users. In addition, one can find other proposals in the literature for inputs based on direct and indirect observations, such as the human behavior and psychological factors [38,39], inferred observations, as well as ageing mechanisms [38,39,41,44]. However, these blockchain-based models do not guarantee that false, unfair, or biased reputation data enters into the chain.

The blockchain paradigm, due to its properties of decentralization, immutability, and tamper-proofness, provides the conditions to mitigate the aforementioned limitations present in the current approaches to reputation systems. One fundamental problem is the lack of trust among the entities that manage e-commerce platforms. The blockchain may be the means to archive this desirable trust, in which all participants can share their data in a decentralized, tamper-proof, and transparent way, but it does not solve the problem of the blockchain oracles [26–29]. Finally, there is also the significant challenge of global adoption of that blockchain network.

*(iii)    What are the proposed Blockchain-Based Reputation System Architectures?*

In Table 2, we present the list of the proposals that we found in our systematic literature review. As far as we know, these are the most recent and relevant works in the field of blockchain-based reputation systems, in different stages: proposal, model, or prototype.

## 5. Conclusions and Future Work

In this paper, we present and discuss the results of a systematic literature review on blockchain technology applied in user reputation systems following the PRISMA methodology.

We expect to contribute to the enrichment of knowledge in the field, by answering the research questions presented in Section 3.1. As such, our discussion in Section 4 focus on finding out which of the major problems can be addressed by means of a blockchain-based architecture, as well as the new challenges that this approach poses and how they are addressed in each proposal in the studied cases.

According to the literature, two of the major problems in the current reputation systems are a lack of transparency regarding the management of reputation data based on centralized authorities and limitations in identity management. These two problems result from limitations in the reputation model and its system architecture, enabling conditions for successful attacks such as the ones already mentioned.

As far as we know, the two dominant approaches for implementing blockchain-based reputation and recommendation systems are supported on: (1) public blockchains, typically the Ethereum, and (2) permissioned blockchains. In the case of public networks, the reputation systems follow an economic viability model, in which it is not viable to perform fake/unfair reviews, Sybil-based attacks, or clean the user's reputation by creating a new identity. Dishonest users are penalized and honest users are rewarded. A user's identity is linked to the blockchain address of the user. The permission-based approaches are based on consortiums of e-commerce platforms, to which a new member must be accepted and follow the imposed rules. Regarding identity management, the consortium may include a central identity management authority.

A blockchain ensures the transparency of data in the chain, and enables means for improving the effectiveness of the mechanisms of reputation score calculation, but the problem of the oracles that enable fake, unfair, and biased feedback to enter the chain remain to be unsolved.

In future work, we intend to research both public-based and permissioned-based blockchain approaches in order to propose a model for deriving user reputation.

**Author Contributions:** Conceptualization, M.J.A.G., R.H.P.; methodology, M.J.A.G., R.H.P.; formal analysis M.J.A.G., R.H.P.; investigation, M.J.A.G., R.H.P., M.A.G.M.C.; data curation, M.A.G.M.C.; writing—M.A.G.M.C.; writing—review and editing, M.J.A.G., R.H.P.; supervision, M.J.A.G., R.H.P.; project administration, R.H.P.; funding acquisition, M.J.A.G., R.H.P. All authors have read and agreed to the published version of the manuscript.

**Funding:** This work was funded by the "NORTE-01-0145-FEDER-000044" project, supported by Northern Portugal Regional Operational Programme (Norte2020), under the Portugal 2020 Partnership Agreement, through the European Regional Development Fund (FEDER).

**Conflicts of Interest:** The authors declare no conflict of interest.

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
