# Peer review of "User Reputation on E-Commerce: Blockchain-Based Approaches"

_jcp, doi:10.3390/jcp2040046_

Round 1
Reviewer 1 Report
User Reputation on e-commerce – Blockchain-Based Approach
Dear Authors.
Thanks for submitting your paper to the Journal of Cybersecurity and Privacy.
The paper you are submitting displays a review of selected articles concerning blockchain applications in e-commerce. While the paper is interesting, there are some points in my opinion that needs more elaboration. Please consider the following suggestions:
1) English needs a thorough revision. Although I am not a native myself it was evident that the academic delivery was not ready for a journal publication. Therefore I strongly recommend specialized English proofreading and editing.
2) The methodology is quite unusual for a literature review. The article selection is ok, but the article review is a bit unprofessional in my opinion. It seems a collection of reviews of a single article and is not a critical overview of academic research. The review should be organized and distinguished by specific topics and not by articles. I have seen this before but in a very low-level journal. So I am sure this type of delivery should be avoided. I suggest organizing themes of discussion in the introduction and then reviewing the retrieved articles according to these themes.
3) The consideration of blockchain as a panacea for all problems is a bit too hyped. I believe and I hope that now it is almost clear that blockchain cannot solve alone the problems encountered by centralized services since blockchain is itself centralized and suffers more or less the same problems. Because if the database Is decentralized, the data service is still centralized, so there is no reason to believe blockchain data to be more reliable. Those problems and limitations should be at least mentioned in the introduction-discussion section.
I could not find papers that discuss this issue in e-commerce specifically but you can have a look at these for your reference.
Caldarelli, G. (2020). Understanding the Blockchain Oracle Problem : A Call for Action. Information, 11(11). https://doi.org/10.3390/info11110509
Frankenreiter, J. (2019). The Limits of Smart Contracts. Journal of Institutional and Theoretical Economics JITE, 175(1), 149–162. https://doi.org/10.1628/jite-2019-0021
Damjan, M. (2018). The interface between blockchain and the real world. Ragion Pratica, 2018(2), 379–406. https://doi.org/10.1415/91545
Egberts, A. (2017). The Oracle Problem - An Analysis of how Blockchain Oracles Undermine the Advantages of Decentralized Ledger Systems. SSRN Electronic Journal. https://doi.org/10.2139/ssrn.3382343
Although I highlighted a few changes I recommend the author carefully elaborate on these to improve the quality of their manuscript.
Good Luck with your research.
Author Response
Comment |
Revision |
1) English needs a thorough revision. Although I am not a native myself it was evident that the academic delivery was not ready for a journal publication. Therefore I strongly recommend specialized English proofreading and editing.
|
The English was improved. |
2) The methodology is quite unusual for a literature review. The article selection is ok, but the article review is a bit unprofessional in my opinion. It seems a collection of reviews of a single article and is not a critical overview of academic research. The review should be organized and distinguished by specific topics and not by articles. I have seen this before but in a very low-level journal. So I am sure this type of delivery should be avoided. I suggest organizing themes of discussion in the introduction and then reviewing the retrieved articles according to these themes.
|
The discussion of the articles was changed. First, we present the results of the systematic literature review (methodology approach and results - sections 3). In section 4, we present an analysis in-depth study of the studies, listed in section 3. |
3) The consideration of blockchain as a panacea for all problems is a bit too hyped. I believe and I hope that now it is almost clear that blockchain cannot solve alone the problems encountered by centralized services since blockchain is itself centralized and suffers more or less the same problems. Because if the database Is decentralized, the data service is still centralized, so there is no reason to believe blockchain data to be more reliable. Those problems and limitations should be at least mentioned in the introduction-discussion section.
|
Yes, the blockchain is not a silver bullet for all problems, on the contrary, it introduces new challenges. In the previous version of the paper, we mentioned the problem of false, unfair and biased reputation data that is provided by the users (sellers/buyers) to the e-commerce platforms, which ends at the blockchain, without making a reference to the blockchain oracles problem. In the revised paper these limitations are presented in the introduction and discussed in section 5 (discussion), as well as in the conclusions. The proposed papers were analysed and references were added. |
Reviewer 2 Report
This paper uses a systematic literature review in the field of blockchain-based reputation systems for e-commerce. However, the paper needs major corrections as per below suggestion:
1. high similarity (27%) with internet sources - reduce the similarity to below 20%
2. abstract - add finding
3. discuss findings in Tables 1 and 2.
4. conclusion - add future works
5. need proofread as many grammatical errors found in the paper
Author Response
Comment |
revision |
1. high similarity (27%) with internet sources reduce the similarity to below 20% |
Text was rewritten |
2. abstract - add finding |
Done |
3. discuss findings in Tables 1 and 2. |
We introduced a new section: 5.5 – Research outcomes |
4. conclusion - add future works |
Done |
5. need proofread as many grammatical errors found in the paper |
Done |
Round 2
Reviewer 1 Report
Dear Authors.
Thanks for submitting the revised version of your manuscript.
I see there are improvements to the previous version.
However, I still see that the review is organized in a way that I am not familiar with. I can't say it's necessarily wrong to provide a summary of papers and then their discussion but again is not what I am used to reading in SLRs.
I would have preferred if organized differently with a declaration of the topic under analysis in the introduction and then a discussion on these topics utilizing data from the retrieved papers (without the summary of every paper). This will make the paper shorter and more readable. Findings would be also more clear to readers.
Because as you can imagine, the paper is equally valuable without the summary of each paper.
At least a reduction of summaries and a small enhancement to the finding section?
English is improved but can be better.
Good luck with your manuscript.
Author Response
Response to reviews
Thank you very much for your contribution to improving the article.
Comment |
Revision |
“ I still see that the review is organized in a way that I am not familiar with. I can't say it's necessarily wrong to provide a summary of papers and then their discussion but again is not what I am used to reading in SLRs. |
Section 4 was removed. We added another column to the table. This column contains a short description of the article. |
English is improved but can be better. |
The English was revised |
Reviewer 2 Report
This paper already improves based on reviewer comments. Thus, I recommended that the paper enough for journal publication.
Author Response

(The authors gave the same response as above.)
